# Combined Screw and Wedge Dislocations

Mikhail O. Katanaev [1],*  and Alexander V. Mark [2]

1 Steklov Mathematical Institute, ul. Gubkina, 8, Moscow 119991, Russia

2 Moscow Automobile and Road Construction State, Technical University (MADI), Leningradsky Prospect, 64, Moscow 125319, Russia; am-83-45@yandex.ru

* Correspondence: katanaev@mi-ras.ru

**Abstract:** Elastic media with defects are considered manifold with nontrivial Riemann–Cartan geometry in the geometric theory of defects. We obtain the solution of three-dimensional Euclidean general relativity equations with an arbitrary number of linear parallel sources. It describes elastic media with parallel combined wedge and screw dislocations.

**Keywords:** geometric theory of defects; dislocation; three-dimensional gravity



## 1. Introduction

The majority of the physical properties of real solid bodies depend on defects distributed inside media. Therefore, the description of elastic media with dislocations, which are the most common defects, is of great importance for applications.

One of the most promising approach to this problem is based on Riemann–Cartan geometry. The idea to relate linear dislocations to torsion tensor goes back to the fifties of the last century [1–5]. This approach has been successfully developed (see [6–16]), and is often called gauge theory of dislocations.

There are other types of defects in media with spin structure, for example, in ferromagnets and liquid crystals, called disclinations [17]. The presence of disclinations may be also interpreted as the appearance of nontrivial geometry on a manifold. The gauge approach based on rotation groups for description of disclinations is also used, for example, in [18–20].

A geometric approach for the description of both types of defects from a unique point of view was proposed in [21]. In this approach, contrary to others, the only independent variables are the vielbein and $\mathbb{SO}(3)$ connection describing static distribution of dislocations and disclinations. The curvature and torsion tensors have physical meaning as surface densities of Frank and Burgers vectors, respectively. Equilibrium equations for defects are assumed to be invariant under general coordinate transformations and local $\mathbb{SO}(3)$ rotations. Since any solution to field equations is defined up to local transformations, we have to impose gauge conditions. The elastic gauge for the vielbein [22] and Lorentz gauge for the $\mathbb{SO}(3)$ connection [23] connect the geometric theory of defects with ordinary linear elasticity theory and the principal chiral $\mathbb{SO}(3)$ field. The possible relation of geometric theory of defects and nonlinear elasticity is discussed in [24].

The geometric theory of defects has a great advantage compared to many other approaches. It is well suited for the description of single defects as well as their continuous distribution.

The presence of defects results in nontrivial Riemann–Cartan geometry. Therefore, we have to replace the Euclidean metric by the Riemannian metric to describe the physical properties of media with defects. For example, the scattering of phonons on wedge dislocations was analyzed in [25,26]. It is also shown that the presence of defects essentially changes many properties of real solids (see, for example, [27–34]).

The traditional theory of single defects within the elasticity theory [4] uses the displacement vector field as the main independent variable with appropriate boundary conditions. In the geometric theory of single dislocations, the only independent variable is the vielbein field satisfying second-order equilibrium equations. The displacement vector field is reconstructed afterwards from the vielbein field in those domains where the geometry is Euclidean (curvature and torsion vanish). In the present paper, we follow this way of thought exactly. First, we solve three-dimensional Einstein equations and find the metric. Then we find the transformation of coordinates, which brings the obtained solution to the Euclidean metric outside the dislocation sources, the displacement vector field being defined by the coordinate transformation.

It is understood that for the description of the continuous distribution of defects, one has to introduce a new variable instead of the displacement vector field [1–5] which is, in fact, the triad or metric field. The geometric theory of defects proposes equations for this new variable based on Euclidean three-dimensional gravity models in Riemann–Cartan spacetimes.

The geometric theory of defects is still under construction, and expression for the free energy for real bodies is not fixed. Originally, the three-dimensional Euclidean version of Riemann–Cartan quadratic gravity was proposed in [21]. If disclinations are absent, then equations for vielbein are reduced to pure Euclidean general relativity. Another interesting possibility is to use the Chern–Simons action for $\mathbb{SO}(3)$ connection [35,36]. Note that in the absence of elastic stresses, the metric is Euclidean, and famous 't Hooft–Polyakov monopole solutions have a straightforward physical interpretation in the geometric theory of defects [37] and may be observed, hopefully, in real solids [38].

Last year, great attention was paid to analogue gravity [39], which investigates analogues of gravitational field phenomena in condensed matter physics. In particular, the behavior of fields in curved spacetime can be studied in the laboratory (see [40,41] for review). In analogue gravity, nontrivial metrics arise as the consequence of field equations for condensed matter, which differ from gravity equations. On the contrary, in the geometric theory of defects, gravity equations themselves are used for the description of defects distribution in matter.

We are not aware of experimental confirmation of the geometric theory of defects, but there have been some attempts to use these ideas in explaining the known properties of real solids (see, for example, [42–44]).

In the present paper, we consider Euclidean three-dimensional general relativity with linear sources as the equilibrium equations for the vielbein. The solution of the field equations are obtained in close analogy with the Lorentzian version of point particles in three-dimensional general relativity [45]. We show that this solution describes the arbitrary distribution of parallel combined wedge and screw dislocations. It seems to be new in solid-state physics.

## 2. Notation

In the geometric theory of defects, dislocations and disclinations correspond to nontrivial torsion and curvature, respectively. If disclinations are absent, then the curvature vanishes, and we are left with teleparallel space. It is well known that teleparallel gravity is equivalent to general relativity (see, for example, [46]). So, we shall use general relativity as a more familiar background. In this way, we obtain the metric as the solution of Euclidean three-dimensional Einstein equations. Afterwards, the vielbein and torsion can be reconstructed, but this is not needed in what follows.

We consider three-dimensional Euclidean space with Cartesian coordinates $(x^\alpha) \in \mathbb{R}^3$, $\alpha = 1, 2, 3$. Let there be a second positive definite metric $g_{\alpha\beta}$ describing the distribution of linear dislocations in the geometric theory of defects. It defines Christoffel's symbols as usual:

$$\Gamma_{\alpha\beta\gamma} := \frac{1}{2}(\partial_\alpha g_{\beta\gamma} + \partial_\beta g_{\alpha\gamma} - \partial_\gamma g_{\alpha\beta}).$$

The curvature and Ricci tensors and scalar curvature are defined as

$$R_{\alpha\beta\gamma}{}^{\delta} := \partial_\alpha \Gamma_{\beta\gamma}{}^{\delta} - \partial_\beta \Gamma_{\alpha\gamma}{}^{\delta} - \Gamma_{\alpha\gamma}{}^{\epsilon} \Gamma_{\beta\epsilon}{}^{\delta} + \Gamma_{\beta\gamma}{}^{\epsilon} \Gamma_{\alpha\epsilon}{}^{\delta},$$

$$R_{\alpha\gamma} := R_{\alpha\beta\gamma}{}^{\beta}, \qquad R := g^{\alpha\gamma} R_{\alpha\gamma},$$

where the indices are lowered and raised by the metric $g_{\alpha\beta}$ and its inverse $g^{\alpha\beta}$.

Let there be N linear, possibly curved, dislocations with axes $\left(q_I^\alpha(\tau)\right) \in \mathbb{R}^3$, $\tau \in \mathbb{R}$, where index $I = 1, \dots, N$ enumerates the dislocations. The free energy expression for the static distribution of N linear dislocations is chosen to be

$$S := -\kappa \int dx \sqrt{|g|} R + \sum_{I=1}^{N} m_I \int d\tau \sqrt{\dot{q}_I^\alpha \dot{q}_I^\beta g_{\alpha\beta}}, \tag{1}$$

where $\dot{q}_I := \partial_\tau q_I$; $\kappa \in \mathbb{R}$ is the coupling constant, and $m_I \in \mathbb{R}$ is the "strength" of the I-th dislocation. We assume that the dislocation axes do not intersect between themselves and may be parameterized by points on the $x^3$ axis, i.e., $\dot{q}_I^3 \neq 0$. Then the equations of equilibrium for the arbitrary distribution of linear dislocations are

$$R_{\alpha\beta} - \frac{1}{2} g_{\alpha\beta} R = -\frac{1}{2} T_{\alpha\beta}, \tag{2}$$

$$\ddot{q}_I^\alpha + \Gamma_{\beta\gamma}{}^{\alpha} \dot{q}_I^\beta \dot{q}_I^\gamma = 0, \tag{3}$$

where

$$T^{\alpha\beta} := \frac{1}{\sqrt{|g|}} \sum_I \frac{m_I \dot{q}_I^\alpha \dot{q}_I^\beta}{\dot{q}_I^3} \delta(\boldsymbol{x} - \boldsymbol{q}_I),$$

$$\delta(\boldsymbol{x} - \boldsymbol{q}_I) := \delta(x^1 - q_I^1) \delta(x^2 - q_I^2)$$

is the two-dimensional $\delta$-function with support on points where the I-th dislocation axis intersects the $x^1, x^2$ plane.

So the arbitrary static distribution of linear dislocations is described by the Euclidean version of *n* point particles in three-dimensional general relativity. Outside the dislocation axes, the space is flat because the total curvature tensor in three dimensions is defined uniquely by its Ricci tensor which vanishes outside sources. The curvature is singular on the dislocation lines. In this sense, the space with dislocations becomes curved. In equilibrium, dislocations axes are located along geodesic lines in three-dimensional Riemannian space, and they curve the initial Euclidean space, describing elastic media with dislocations. To obtain a connection with the ordinary elasticity theory, we have to impose an elastic gauge containing Lame coefficients [22]. Unfortunately, the solution of field Equation (2) in an elastic gauge is not the easiest way. Therefore, we impose another gauge in the next section to obtain the exact solution. The problem of rewriting this solution in an elastic gauge is purely technical and left for future analysis.

## 3. Einstein Equations

We assume that all dislocations are straight and parallel to the $x^3$ axis. Moreover, dislocations are supposed to be homogeneous along the $x^3$ axis, i.e., there is the translational symmetry $x^3 \mapsto x^3 +$ const. Then the metric can be parameterized as

$$g_{\alpha\beta} = \begin{pmatrix} h^2 & -h^2 \omega_\nu \\ -h^2 \omega_\mu & \bar{g}_{\mu\nu} + h^2 \omega_\mu \omega_\nu \end{pmatrix} \quad \Leftrightarrow \quad g^{\alpha\beta} = \begin{pmatrix} \dfrac{1}{h^2} + \omega^2 & \omega^\nu \\ \omega^\mu & \bar{g}^{\mu\nu} \end{pmatrix}, \tag{4}$$

where $h(x) > 0$, $\omega_\mu(x)$, and $\bar{g}_{\mu\nu}(x)$ are some unknown functions on two coordinates $x := (x^\mu) \in \mathbb{R}^2$, $\mu := 1, 2$ such that $\det \bar{g}_{\mu\nu} \neq 0$. The corresponding interval is

$$ds^2 = h^2 \left(dx^3 - \omega_\mu dx^\mu\right)^2 + \bar{g}_{\mu\nu} dx^\mu dx^\nu. \tag{5}$$

There is the identity

$$\det g_{\alpha\beta} = h^2 \det \bar{g}_{\mu\nu}.$$

Thus, the whole problem is reduced to a two-dimensional one, and indices from the middle of Greek alphabet will take only two values $\mu, \nu, \ldots = 1, 2$.

Parameterization (4) is possible in any dimensions and coincides with the Arnowitt–Deser–Misner parameterization for the *inverse* metric.

A general solution of field equations for $\omega_\mu \equiv 0$ in the context of the geometric theory of defects was found in [21]. It describes the arbitrary distribution of parallel wedge dislocations. Therefore, we consider now the case $\omega_\mu \neq 0$. In three-dimensional general relativity, covector components $\omega_\mu$ are related to spins of point particles [47], whereas they correspond to screw dislocations in the geometric theory of defects.

Christoffel's symbols with lower indices for metric (4) are

$$\begin{aligned}
\Gamma_{333} &= 0, \qquad \Gamma_{33\mu} = -h\partial_\mu h, \qquad \Gamma_{3\mu 3} = h\partial_\mu h, \\
\Gamma_{3\mu\nu} &= -h\partial_\mu h\omega_\nu + h\partial_\nu h\omega_\mu - \frac{1}{2}h^2 F_{\mu\nu}, \\
\Gamma_{\mu\nu 3} &= -h\partial_\mu h\omega_\nu - h\partial_\nu h\omega_\mu - \frac{1}{2}h^2(\partial_\mu\omega_\nu + \partial_\nu\omega_\mu), \\
\Gamma_{\mu\nu\rho} &= \bar{\Gamma}_{\mu\nu\rho} + h(\partial_\mu h\omega_\nu + \partial_\nu h\omega_\mu)\omega_\rho - h\partial_\rho h\omega_\mu\omega_\nu \\
&\quad + \frac{1}{2}h^2(\partial_\mu\omega_\nu\omega_\rho + \partial_\nu\omega_\mu\omega_\rho + \omega_\mu F_{\nu\rho} + \omega_\nu F_{\mu\rho}),
\end{aligned} \tag{6}$$

where overlined Christoffel's symbols $\bar{\Gamma}_{\mu\nu\rho}$ are built for two-dimensional metric $\bar{g}_{\mu\nu}$, and

$$F_{\mu\nu} := \partial_\mu\omega_\nu - \partial_\nu\omega_\mu$$

is the "field strength" for covector field $\omega_\mu$. Christoffel's symbols with one upper index have the form

$$\begin{aligned}
\Gamma_{33}{}^3 &= -h\partial_\mu h\omega^\mu, \\
\Gamma_{33}{}^\mu &= -h\partial_\nu \bar{g}^{\nu\mu}, \\
\Gamma_{3\mu}{}^3 &= \frac{\partial_\mu h}{h} + h\partial_\nu h\omega^\nu\omega_\mu - \frac{1}{2}h^2 F_{\mu\nu}\omega^\nu, \\
\Gamma_{3\mu}{}^\nu &= h\partial_\rho h\bar{g}^{\rho\nu}\omega_\mu - \frac{1}{2}h^2 F_\mu{}^\nu, \\
\Gamma_{\mu\nu}{}^3 &= -\frac{1}{h}(\partial_\mu h\omega_\nu + \partial_\nu h\omega_\mu) - \frac{1}{2}(\bar{\nabla}_\mu\omega_\nu + \bar{\nabla}_\nu\omega_\mu) - h\partial_\rho h\omega^\rho\omega_\mu\omega_\nu \\
&\quad + \frac{1}{2}h^2(\omega_\mu F_{\nu\rho} + \omega_\nu F_{\mu\rho})\omega^\rho, \\
\Gamma_{\mu\nu}{}^\rho &= \bar{\Gamma}_{\mu\nu}{}^\rho - h\partial_\sigma h\bar{g}^{\sigma\rho}\omega_\mu\omega_\nu + \frac{1}{2}h^2(\omega_\mu F_\nu{}^\rho + \omega_\nu F_\mu{}^\rho),
\end{aligned} \tag{7}$$

where the raising of the indices is performed by using the inverse two-dimensional metric $\bar{g}^{\mu\nu}$ and $\bar{\nabla}_\mu$ denotes the two-dimensional covariant derivative for metric $\bar{g}_{\mu\nu}$.

Straightforward calculations yield the curvature tensor. Its linear independent nonzero components are:

$$R_{3\mu 3\nu} = h\bar{\nabla}_\mu \bar{\nabla}_\nu h - \frac{1}{4}h^4 F_\mu{}^\rho F_{\nu\rho},$$

$$R_{3\mu\nu\rho} = h\bar{\nabla}_\mu \bar{\nabla}_\nu h\omega_\rho - h\bar{\nabla}_\mu \bar{\nabla}_\rho h\omega_\nu + \frac{h}{2}\partial_\nu h F_{\mu\rho} - \frac{h}{2}\partial_\rho h F_{\mu\nu} + h\partial_\mu h F_{\nu\rho}$$

$$+ \frac{h^2}{2}\bar{\nabla}_\mu F_{\nu\rho} - \frac{h^4}{4}F_\mu{}^\sigma F_{\nu\sigma}\omega_\rho + \frac{h^4}{4}F_\mu{}^\sigma F_{\rho\sigma}\omega_\nu,$$

$$R_{\mu\nu\rho\sigma} = \bar{R}_{\mu\nu\rho\sigma} + h\partial_\rho h\omega_\sigma F_{\mu\nu} - h\partial_\sigma h\omega_\rho F_{\mu\nu} + h\partial_\mu h\omega_\nu F_{\rho\sigma} - h\partial_\nu h\omega_\mu F_{\rho\sigma}$$

$$+ h\bar{\nabla}_\mu \bar{\nabla}_\rho h\omega_\nu\omega_\sigma - h\bar{\nabla}_\mu \bar{\nabla}_\sigma h\omega_\nu\omega_\rho - h\bar{\nabla}_\nu \bar{\nabla}_\rho h\omega_\mu\omega_\sigma + h\bar{\nabla}_\nu \bar{\nabla}_\sigma h\omega_\mu\omega_\rho$$

$$+ \frac{h}{2}\partial_\rho h\omega_\nu F_{\mu\sigma} - \frac{h}{2}\partial_\sigma h\omega_\nu F_{\mu\rho} - \frac{h}{2}\partial_\rho h\omega_\mu F_{\nu\sigma} + \frac{h}{2}\partial_\sigma h\omega_\mu F_{\nu\rho}$$

$$+ \frac{h}{2}\partial_\nu h\omega_\sigma F_{\mu\rho} - \frac{h}{2}\partial_\nu h\omega_\rho F_{\mu\sigma} - \frac{h}{2}\partial_\mu h\omega_\sigma F_{\nu\rho} + \frac{h}{2}\partial_\mu h\omega_\rho F_{\nu\sigma}$$

$$+ \frac{h^2}{2}\omega_\sigma \bar{\nabla}_\rho F_{\mu\nu} - \frac{h^2}{2}\omega_\rho \bar{\nabla}_\sigma F_{\mu\nu} + \frac{h^2}{2}\omega_\nu \bar{\nabla}_\mu F_{\rho\sigma} - \frac{h^2}{2}\omega_\mu \bar{\nabla}_\nu F_{\rho\sigma}$$

$$+ \frac{h^2}{2}F_{\mu\nu}F_{\rho\sigma} + \frac{h^2}{4}F_{\mu\rho}F_{\nu\sigma} - \frac{h^2}{4}F_{\nu\rho}F_{\mu\sigma}$$

$$+ \frac{h^4}{4}\omega_\mu\omega_\sigma F_\nu{}^\lambda F_{\rho\lambda} - \frac{h^4}{4}\omega_\mu\omega_\rho F_\nu{}^\lambda F_{\sigma\lambda} - \frac{h^4}{4}\omega_\nu\omega_\sigma F_\mu{}^\lambda F_{\rho\lambda} + \frac{h^4}{4}\omega_\nu\omega_\rho F_\mu{}^\lambda F_{\sigma\lambda}. \tag{8}$$

The Ricci tensor components become:

$$R_{33} = h\bar{\triangle}h - \frac{h^4}{4}F^2,$$

$$R_{3\mu} = -\left(h\bar{\triangle}h - \frac{h^4}{4}F^2\right)\omega_\mu + \frac{1}{2h}\bar{\nabla}_\nu(h^3 F_\mu{}^\nu)$$

$$= -R_{33}\omega_\mu + \frac{1}{2h}\bar{\nabla}_\nu(h^3 F_\mu{}^\nu),$$

$$R_{\mu\rho} = \bar{R}_{\mu\rho} + \frac{1}{h}\bar{\nabla}_\mu \bar{\nabla}_\rho h + \left(h\bar{\triangle}h - \frac{h^4}{4}F^2\right)\omega_\mu\omega_\rho \tag{9}$$

$$- \frac{1}{2h}\omega_\rho \bar{\nabla}_\nu(h^3 F_\mu{}^\nu) - \frac{1}{2h}\omega_\mu \bar{\nabla}_\nu(h^3 F_\rho{}^\nu) + \frac{h^2}{2}F_{\mu\nu}F_\rho{}^\nu$$

$$= \bar{R}_{\mu\rho} + \frac{1}{h}\bar{\nabla}_\mu \bar{\nabla}_\rho h + \frac{h^2}{2}F_{\mu\nu}F_\rho{}^\nu - R_{33}\omega_\mu\omega_\rho - R_{3\mu}\omega_\rho - R_{3\rho}\omega_\mu,$$

where $\bar{\triangle} := \bar{g}^{\mu\nu}\bar{\nabla}_\mu \bar{\nabla}_\nu$ is the two-dimensional Laplace–Beltrami operator and $F^2 := F^{\mu\nu}F_{\mu\nu}$. The scalar curvature is

$$R = \bar{R} + \frac{2}{h}\triangle h + \frac{h^2}{4}F^2. \tag{10}$$

Now, we compute the Einstein tensor $G_{\alpha\beta} := R_{\alpha\beta} - \frac{1}{2}g_{\alpha\beta}R$:

$$G_{33} = -\frac{h^2}{2}\bar{R} - \frac{3}{8}h^4 F^2,$$

$$G_{3\mu} = \frac{1}{2h}\bar{\nabla}_\nu(h^3 F_\mu{}^\nu) - G_{33}\omega_\mu, \tag{11}$$

$$G_{\mu\rho} = \bar{G}_{\mu\rho} + \frac{1}{h}\bar{\nabla}_\mu \bar{\nabla}_\rho h - \frac{1}{h}\bar{\triangle}h\bar{g}_{\mu\rho} + \frac{h^2}{2}F_{\mu\nu}F_\rho{}^\nu - \frac{h^2}{8}F^2\bar{g}_{\mu\rho} - G_{33}\omega_\mu\omega_\rho - G_{3\mu}\omega_\rho - G_{3\rho}\omega_\mu.$$

The above formulae are valid in arbitrary dimensions. In three dimensions, there are simplifications. It is well known that Einstein tensor $\bar{G}_{\mu\nu}$ for metric $\bar{g}_{\mu\nu}$ vanishes in two dimensions. Moreover,

$$F_{\mu\nu} = \varepsilon_{\mu\nu}(*F) \qquad \Leftrightarrow \qquad *F := \frac{1}{2}F_{\mu\nu}\varepsilon^{\mu\nu},$$

where $\varepsilon_{\mu\nu}$ is the totally antisymmetric second-rank tensor, $\varepsilon_{12} = 1$. Therefore,

$$F_{\mu\nu}F_{\rho}{}^{\nu} = \bar{g}_{\mu\rho}(*F)^2, \qquad F^2 = 2(*F)^2$$

and

$$F_{\mu\nu}F_{\rho}{}^{\nu} - \frac{1}{2}\bar{g}_{\mu\rho}F^2 \equiv 0.$$

In what follows, we shall need the trace and traceless parts of the two-dimensional Einstein tensor:

$$G := \bar{g}^{\mu\rho}G_{\mu\rho} = -\frac{1}{h}\bar{\triangle}h + \frac{1}{4}h^2F^2 - G_{33}\omega^2 - 2G_{3\mu}\omega^{\mu}, \tag{12}$$

$$G_{\mu\rho} - \frac{1}{2}\bar{g}_{\mu\rho}G = \frac{1}{h}\left(\bar{\nabla}_{\mu}\bar{\nabla}_{\nu}h - \frac{1}{2}\bar{g}_{\mu\rho}\bar{\triangle}h\right) - G_{33}\left(\omega_{\mu}\omega_{\rho} - \frac{1}{2}\bar{g}_{\mu\rho}\omega^2\right)$$
$$- G_{3\mu}\omega_{\rho} - G_{0\rho}\omega_{\mu} + \bar{g}_{\mu\rho}G_{3\nu}\omega^{\nu}. \tag{13}$$

Now we are in a position to solve the Einstein equations.

## 4. Solution of Einstein Equations

We solve now Einstein equations $G_{\alpha\beta} = 0$ outside the dislocation axes for the metric of the general form (4). Here, we follow closely the solution in three-dimensional general relativity [45].

To solve the field equations, we fix the gauge as follows. The translational symmetry

$$x^3 \mapsto x'^3 := x^3 - f(\boldsymbol{x}), \qquad \boldsymbol{x} \mapsto \boldsymbol{x}' := \boldsymbol{x},$$

where $f(\boldsymbol{x})$ is some function of the first two coordinates, does not change metric (4):

$$ds^2 = h^2(dx^3 - \omega_{\mu}dx^{\mu})^2 + \bar{g}_{\mu\nu}dx^{\mu}dx^{\nu}$$
$$= h^2(dx'^3 - \omega'_{\mu}dx^{\mu})^2 + \bar{g}_{\mu\nu}dx^{\mu}dx^{\nu},$$

if the covector field transforms as

$$\omega'_{\mu} = \omega_{\mu} - \partial_{\mu}f.$$

The covariant derivative of the last equation yields equality

$$\bar{\nabla}_{\mu}\omega'^{\mu} = \bar{\nabla}_{\mu}\omega^{\mu} - \bar{\triangle}f.$$

The two-dimensional Laplace–Beltrami equation for $f$,

$$\bar{\triangle}f = \bar{\nabla}_{\mu}\omega^{\mu},$$

has a solution for a sufficiently large class of the right-hand sides. It means that the gauge

$$\bar{\nabla}_{\mu}\omega^{\mu} = 0, \tag{14}$$

is admissible at least locally. It is called the Lorentz gauge by analogy with electrodynamics.

Now, we choose the conformal gauge on sections $x^3 = $ const:

$$\bar{g}_{\mu\nu} = e^{2\phi}\delta_{\mu\nu}, \tag{15}$$

where $\phi(x)$ is some function and $\delta_{\mu\nu} := \mathrm{diag}\,(++)$ is the Euclidean metric.

So, we fix the gauge by Equations (14) and (15) and *use the Euclidean two-dimensional metric $\delta_{\mu\nu}$ to raise and lower the Greek indices*.

Christoffel's symbols in the conformal gauge are

$$\bar{\Gamma}_{\mu\nu}{}^{\rho} = \partial_\mu\phi\,\delta_\nu^\rho + \partial_\nu\phi\,\delta_\mu^\rho - \partial^\rho\phi\,\delta_{\mu\nu}, \qquad \partial^\rho := \delta^{\rho\sigma}\partial_\sigma. \tag{16}$$

Then, the gauge condition (14) takes the form

$$\partial_\mu\omega^\mu := \delta^{\mu\nu}\partial_\mu\omega_\nu = 0, \tag{17}$$

because $\delta^{\mu\nu}\bar{\Gamma}_{\mu\nu}{}^{\rho} = 0$.

We decompose field $\omega_\mu$ into the transverse and longitudinal parts, $\omega_\mu = \omega_\mu^{\mathrm{T}} + \omega_\mu^{\mathrm{L}}$. For the zero boundary condition at infinity $\omega_\mu^{\mathrm{L}}\big|_{|x|=\infty} = 0$, gauge condition (17) means the absence of the longitudinal part $\omega_\mu^{\mathrm{L}} = 0$. Therefore,

$$\omega_\mu = \omega_\mu^{\mathrm{T}} = \varepsilon_\mu{}^\nu\partial_\nu\varphi, \qquad \varepsilon_\mu{}^\nu := \varepsilon_{\mu\rho}\delta^{\rho\nu}, \tag{18}$$

where $\varphi(x)$ is some function on two coordinates. It implies equality

$$F_{\mu\nu} = -\varepsilon_{\mu\nu}\hat{\triangle}\varphi, \tag{19}$$

where $\hat{\triangle} := \delta^{\mu\nu}\partial_\mu\partial_\nu$ is the usual flat Laplacian. Relations

$$\bar{R} = -2\,\mathrm{e}^{-2\phi}\,\hat{\triangle}\phi \qquad \text{and} \qquad F^2 := \bar{g}^{\mu\nu}\bar{g}^{\rho\sigma}F_{\mu\rho}F_{\nu\sigma} = 2\,\mathrm{e}^{-4\phi}(\hat{\triangle}\varphi)^2 \tag{20}$$

can be easily checked.

Equations $G_{33} = 0$ and (11) imply

$$\bar{\nabla}_\nu(h^3 F_\mu{}^\nu) = 0 \qquad \Rightarrow \qquad \partial_\mu(\mathrm{e}^{-2\phi}h^3\hat{\triangle}\varphi) = 0,$$

where Christoffel's symbols (16) and Equation (19) were used. A general solution of this equation has the form

$$\hat{\triangle}\varphi = \lambda h^{-3}\,\mathrm{e}^{2\phi}, \tag{21}$$

where $\lambda \in \mathbb{R}$ is an arbitrary integration constant. Now Equation $G_{33} = 0$ is

$$\hat{\triangle}\phi + \frac{3}{4}\lambda^2 h^{-4}\,\mathrm{e}^{2\phi} = 0. \tag{22}$$

The trace of Einstein Equation (12) and the traceless part (13) yield equations:

$$\hat{\triangle}h - \frac{1}{2}\lambda^2 h^{-3}\,\mathrm{e}^{2\phi} = 0, \tag{23}$$

$$\bar{\nabla}_\mu\bar{\nabla}_\rho h - \frac{1}{2}\bar{g}_{\mu\rho}\hat{\triangle}h = 0. \tag{24}$$

Thus, vacuum Einstein equations $G_{\alpha\beta} = 0$ in the fixed gauge are reduced to an overdetermined system of the five Equations (21)–(24) on three unknown functions $\varphi$, $\phi$, and $h$.

For further analysis, we introduce complex coordinates $z := x^1 + ix^2$ on the plane $x := (x^1, x^2)$. Christoffel's symbols for metric in the conformal gauge (15) have only two nonzero components:

$$\Gamma_{zz}{}^z = 2\partial_z\phi, \qquad \Gamma_{\bar{z}\bar{z}}{}^{\bar{z}} = 2\partial_{\bar{z}}\phi,$$

where the line denotes complex conjugation. Therefore, the following relations hold:

$$\bar{\nabla}_z \bar{\nabla}_z h = \partial^2_{zz} h - 2\partial_z \phi \partial_z h,$$
$$\bar{\nabla}_{\bar{z}} \bar{\nabla}_z h = \partial^2_{\bar{z}z} h,$$
$$\triangle h = 4\, e^{-2\phi} \partial^2_{\bar{z}z} h$$

and their complex conjugates. It can be easily checked that the $\bar{z}z$ component of Equation (24) is identically satisfied, and the $zz$ component produces an equation on function $h(z, \bar{z})$:

$$\partial^2_{zz} h - 2\partial_z \phi \partial_z h = 0. \tag{25}$$

There are two cases: the first one is the most interesting from the physical point of view.

**Case I**: $\partial_z h = 0$. The function $h = h(\bar{z})$ is antiholomorphic. If solutions take only constant values at infinity, then $h = $ const. Without loss of generality, this constant may be set to unity $h = 1$ by stretching the $x^3$ coordinate. Then, Equation (23) holds only for $\lambda = 0$. Afterwards, Equations (21) and (22) are reduced to Laplace equations:

$$\begin{aligned} G_{33} = 0 : & \quad \hat{\triangle} \phi = 0, \\ G_{3\mu} = 0 : & \quad \hat{\triangle} \varphi = 0. \end{aligned} \tag{26}$$

The general solutions of these equations' outside sources are

$$\phi = \sum_{I=1}^{N} \theta_I \ln|x - q_I| + \frac{1}{2} C_1, \qquad C_1 = \text{const},$$
$$\varphi = \sum_{I=1}^{N} b_I \ln|x - q_I| + \frac{1}{2} C_2, \qquad C_2 = \text{const}, \tag{27}$$

where $\theta_I$ and $b_I$, $I = 1, \ldots, N$ are some constants (some of them may be zero). More precisely, Equation (26) holds everywhere on the plane $x \in \mathbb{R}^2$ except fixed points $q_I$, where functions $\phi$ and $\varphi$ have singularities for nonzero constants corresponding to dislocation lines. It can be shown that constants $\theta_I$ are related to coupling constants in the initial action (1) as $\theta_I = 4\pi m_I / \kappa$, and $b_I$ appear as constants of integration. Covector $\omega_\mu$ is

$$\omega_\mu = \varepsilon_\mu{}^\nu \partial_\nu \varphi = \sum_I b_I \frac{\varepsilon_{\mu\nu} x^\nu}{|x - q_I|^2}.$$

Thus, we obtained a general solution of field equations in case I:

$$ds^2 = \left( dx^3 - \sum_I b_I \frac{\varepsilon_{\mu\nu} x^\mu dx^\nu}{|x - q_I|^2} \right)^2 + \prod_I |x - q_I|^{2\theta_I} \delta_{\mu\nu} dx^\mu dx^\nu. \tag{28}$$

Here, we put $C_1 = 1$, which can be always achieved by stretching coordinates $x^\mu$, and integration constant $C_2$ does not affect the metric.

**Case II**: $\partial_z h \neq 0$. Now Equation (25) can be divided by $\partial_z h$,

$$\frac{\partial^2_{zz} h}{\partial_z h} = 2\partial_z \phi, \tag{29}$$

and integrated:

$$\partial_z h = \mu(\bar{z})\, e^{2\phi}, \tag{30}$$

where $\mu(\bar{z})$ is an arbitrary nonzero antiholomorphic function because case I arises for $\mu = 0$. Let us perform conformal transformation $z \mapsto Z(z)$ defined by the equation

$$dz := \bar{\mu}(z) dZ,$$

where the line denotes a complex conjugate. Then, Equation (30) takes the form

$$\frac{\partial h}{\partial Z} = \mu(\bar{z})\bar{\mu}(z)\, e^{2\phi} =: \frac{1}{2}\, e^{2\Phi}, \tag{31}$$

where $\Phi(z, \bar{z})$ is real.

Note that isotropic coordinates (15) are defined up to conformal transformations on the complex plane. Therefore, function $\mu(\bar{z})$ describes this arbitrariness. In other words, function $\mu(\bar{z})$, which arose during the integration of Equation (29), can be set to unity without loss of generality.

The complex conjugate to Equation (31) together with the reality of functions $h$ and $\Phi$ implies that functions $h = h(X)$ and $\Phi = \Phi(X)$ depend only on one coordinate

$$X = \frac{1}{2}(Z + \bar{Z}), \qquad Z := X + iY.$$

Then, Equation (31) takes the form

$$\frac{dh}{dX} = e^{2\Phi}. \tag{32}$$

Now, Equations (22) and (23) imply the system of ordinary differential equations for functions $h$ and $\Phi$:

$$\frac{d^2h}{dX^2} - \frac{1}{2}\lambda^2 h^{-3}\, e^{2\Phi} = 0, \tag{33}$$

$$\frac{d^2\Phi}{dX^2} + \frac{3}{4}\lambda^2 h^{-4}\, e^{2\Phi} = 0. \tag{34}$$

Using Equation (32), Equation (33) is easily integrated

$$\frac{dh}{dX} = -\frac{\lambda^2}{4h^2} + \nu, \tag{35}$$

where $\nu \in \mathbb{R}$ is an integration constant. This equation can be also integrated, but we shall not need it. We proceed with the investigation by writing the metric in coordinates $x^3, h, Y$, which does not require the explicit form of $h = h(X)$.

Note that Equation (32) implies that the right-hand side of Equation (35) must be positive. This restricts possible values of constant $\nu$, which will be analyzed later.

The expression for $\Phi$ follows from Equations (32) and (35)

$$\Phi = \frac{1}{2}\ln\left(-\frac{\lambda^2}{4h^2} + \nu\right) \qquad \Leftrightarrow \qquad e^{2\Phi} = -\frac{\lambda^2}{4h^2} + \nu. \tag{36}$$

Differentiate it with respect to $X$:

$$2\frac{d\Phi}{dX} = \left(-\frac{\lambda^2}{4h^2} + \nu\right)^{-1}\frac{\lambda^2}{2h^3}\frac{dh}{dX} = \frac{\lambda^2}{2h^3},$$

where Equation (35) is used. Next the differentiation on $X$ leads to Equation (34). Thus, the first-order Equation (35) solves the system of Equations (33) and (34) under restriction (32).

Now, we have to find function $\varphi$. It satisfies only one Equation (21), which has the form in new coordinates

$$\triangle\varphi = \lambda h^{-3}\, e^{2\Phi}, \qquad \triangle := \frac{\partial^2}{\partial X^2} + \frac{\partial^2}{\partial Y^2}. \tag{37}$$

A general solution of this equation is equal to the sum of the general solution of the homogeneous Laplace equation $\triangle \varphi = 0$ and some particular solution of the inhomogeneous equation, which is expressed through function $h$ using Equation (33):

$$\varphi = f(X, Y) + \frac{2}{\lambda} h(X), \tag{38}$$

where $f$ is an arbitrary harmonic function $\triangle f = 0$.

Thus, the metric in case II in coordinates $x^3, X, Y$ has the general form

$$ds^2 = h^2 \big(dx^3 - \varepsilon_\mu{}^\nu \partial_\nu \varphi \, dX^\mu\big)^2 + \left(-\frac{\lambda^2}{4h^2} + \nu\right)(dX^2 + dY^2), \tag{39}$$

where functions $h(X)$ and $\varphi(X, Y)$ are given by Equations (35) and (38), respectively.

Metric (39) can be written in a simpler form which does not require the solution of differential Equation (35). Function $h(X)$ depends only on one coordinate, and we go to a new coordinate system $x^3, X, Y \mapsto x^3, h, Y$. To simplify the resulting metric, we redefine the harmonic function entering Equation (38):

$$\varphi \mapsto \varphi = f(X, Y) + \frac{2}{\lambda}\big[h(X) - \nu X\big]. \tag{40}$$

Then

$$\omega_\mu dx^\mu = \frac{\partial \varphi}{\partial Y} dX - \frac{\partial \varphi}{\partial X} dY = \frac{\partial f}{\partial Y} dX - \frac{\partial f}{\partial X} dY - \frac{\lambda}{2h^2} dY.$$

Substitution of this equality in Equation (39) yields the simpler answer

$$ds^2 = h^2 dT^2 - \lambda \, dT \, dY + \nu \, dY^2 - \left(\frac{\lambda^2}{4h^2} - \nu\right)^{-1} dh^2, \tag{41}$$

where we changed the third coordinate $x^3 \mapsto T$, which is defined by the differential equation

$$dT = dx^3 - \frac{\partial f}{\partial Y} dX + \frac{\partial f}{\partial X} dY. \tag{42}$$

Coordinate $T$ exists at least locally because the second exterior derivative vanishes $d^2 T = 0$ due to the harmonicity of function $f$.

During the solution of field equations, we restrict constant $\nu$

$$-\frac{\lambda^2}{4h^2} + \nu \geq 0. \tag{43}$$

However, metric (41) is defined for all $\nu$ and consequently satisfies vacuum Einstein equations. It is the analytic continuation of the solution for all $\nu \in \mathbb{R}$.

Next, coordinate transformation $Y := y + \frac{\lambda}{2\nu} T$ transforms metric (41) to the diagonal form

$$ds^2 = (h^2 - M)dT^2 + \frac{h^2}{\nu(h^2 - M)} dh^2 + \nu \, dy^2, \qquad M := \frac{\lambda^2}{2\nu}.$$

Now, the two-dimensional part $T, h$ of the metric can be easily transformed to a conformally flat form by substitution of $h \mapsto u$, where

$$\left|\frac{du}{dh}\right| = \frac{h}{\sqrt{\nu(h^2 - M)}}, \qquad \nu > 0.$$

Similar transformations can be performed for $\nu < 0$. Then, the metric becomes

$$ds^2 = \mathrm{e}^u (dT^2 + du^2) + dy^2 \tag{44}$$

up to the rescaling of coordinates and the total constant factor.

It is easily checked that the curvature tensor for this metric vanishes, as it should be. Metric (44) does not depend on constants $\theta_I$, and its physical interpretation within the geometric theory of defects remains unclear.

*Combined Wedge and Screw Dislocations*

In case I, solution (28) of the Einstein equations depends on two sets of constants $\theta_1, \ldots, \theta_N$ and $b_1, \ldots, b_N$. We now give them a physical interpretation in the geometric theory of defects. Consider one straight dislocation located along the $x^3$ axis. Then, metric (28) takes the form

$$ds^2 = \left(dx^3 - b\frac{\varepsilon_{\mu\nu}x^\mu dx^\nu}{x^\rho x_\rho}\right)^2 + (x^\nu x_\nu)^\theta dx^\mu dx_\mu, \qquad \theta, b = \text{const.} \qquad (45)$$

In polar coordinates

$$x^1 := r\cos\varphi, \qquad x^2 := r\sin\varphi,$$

the metric is

$$ds^2 = (dx^3 + bd\varphi)^2 + r^{2\theta}(dr^2 + r^2 d\varphi^2), \qquad (46)$$

or

$$ds^2 = (d\tilde{x}^3)^2 + r^{2\theta}(dr^2 + r^2 d\varphi^2),$$

where $\tilde{x}^3 := x^3 + b\varphi$. If $b = 0$, then the two-dimensional $r, \varphi$ part of metric (46) describes conical singularity on the plane with deficit angle $\theta$.

The dislocation line coincides with the $x^3$ axis. At this axis, $h \equiv 1$ and $\omega_\mu \equiv 0$. Therefore, $\Gamma_{33}{}^3\big|_{x^\mu=0} \equiv 0$, and geodesic Equation (3) are satisfied. So we have the consistent solution of equilibrium equations.

If $b = 0$, then metric (46) describes wedge dislocation. For negative $-1 < \theta < 0$, it is produced by cutting out the wedge of media parallel to the $x^3$ axis with angle $-2\pi\theta$ and symmetrically gluing together both sides of the cut (see Figure 1). For positive $\theta$, the wedge is inserted inside the media.

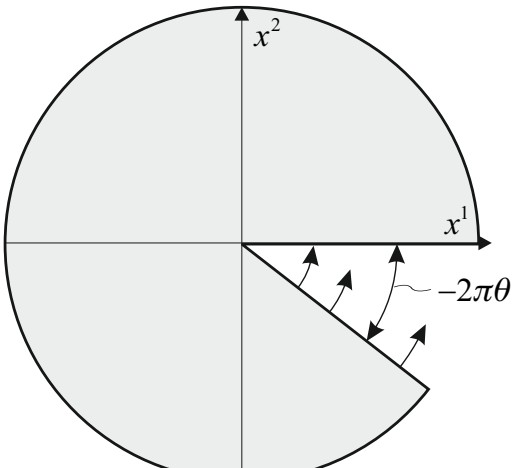

**Figure 1.** The slice $x^3 = \text{const}$ of the media with the wedge dislocation for $-1 < \theta < 0$.

For $\theta = 0$, metric (46) corresponds to screw dislocation with Burgers vector $|\boldsymbol{b}| = b$ shown in Figure 2.

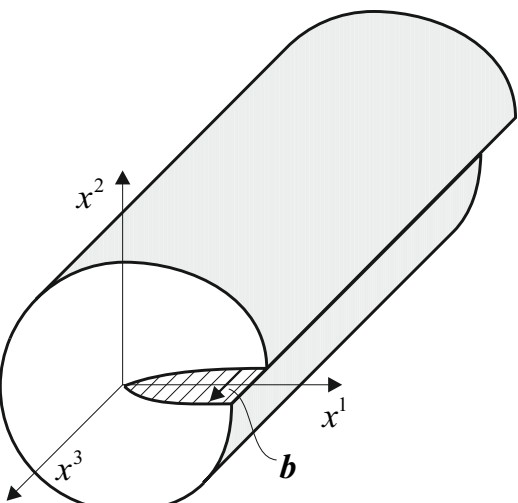

**Figure 2.** The screw dislocation with Burgers vector $|\boldsymbol{b}| = b$.

It is produced as follows. We make the half plane cut in media in the $x^3, x^1$ plane, axis $x^3$ being the edge of the cut. Then, the lower part of the media is moved along the $x^3$ axis, and both sides of the cut are glued together. The magnitude of the displacement that is far enough from the dislocation line $x^3$ is assumed to be constant and called the Burgers vector.

The transformation of coordinates

$$\tilde{x}^3 := x^3 + b\varphi, \quad \tilde{r} := \frac{1}{\gamma} r^\gamma, \quad \tilde{\varphi} := \gamma\varphi, \qquad \gamma := 1 + \theta,$$

brings metric (46) to the Euclidean form

$$ds^2 = (d\tilde{x}^3)^2 + d\tilde{r}^2 + \tilde{r}^2 d\tilde{\varphi}^2.$$

Rewriting this coordinate transformation in Cartesian coordinates, one easily obtains the displacement vector field $u^\alpha(x) := x^\alpha - \tilde{x}^\alpha(x)$. This coordinate transformation is degenerate along the $x^3$ axis. Therefore, the space with dislocation is not Euclidean as a whole.

Thus, metric (46) in case I describes combined wedge and screw dislocations.

## 5. Conclusions

We described an arbitrary distribution of $n$ static straight parallel dislocations with deficit angles $\theta_I$, $I = 1, \ldots, N$ and Burgers vectors $b_I$ within the geometric theory of defects. This problem is hardly to be solved in ordinary elasticity theory because of the very complicated boundary conditions for the displacement vector field. The obtained metric describes the distribution of elastic stresses inside media but does not depend on Lame coefficients and therefore cannot be directly observed. To build the bridge to experiments, the metric should be rewritten in an elastic gauge [22] which depends on Lame coefficients and allows the metric to be attributed to particular elastic media. This problem is technical and left for further study. Anyway, it is important to note that general relativity may help to solve problems in elasticity theory, and sometimes the geometric approach seems to be easier and more general.

**Author Contributions:** Conceptualization, M.O.K.; investigation, M.O.K. and A.V.M.; writing—original draft preparation, M.O.K. and A.V.M.; writing—review and editing, M.O.K. and A.V.M. All authors have read and agreed to the published version of the manuscript.

**Funding:** The work of M.O. Katanaev was performed at the Steklov International Mathematical Center and supported by the Ministry of Science and Higher Education of the Russian Federation (agreement no. 075-15-2022-265).

**Data Availability Statement:** The data presented in this study are available in the article.

**Conflicts of Interest:** The authors declare no conflict of interest.

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
