# Peer review of "Combined Screw and Wedge Dislocations"

_universe, doi:10.3390/universe9120500_

Round 1

Reviewer 1 Report

Comments and Suggestions for Authors

In the manuscript entitled "Combined screw and wedge dislocations", the authors provide through detailed calculations the metric tensor associated to parallel combined wedge and screw dislocations. These line defects are associated to line distributions of curvature and torsion respectively [1], and as such, the adequate framework to deal with them is Riemann-Cartan gravity, as recalled in the introduction.

My major concern is precisely about this point: section 2 recalls the basics of Einsteinian 3D gravity, with the usual Levi-Civita connection (Christoffel's symbols) and the equations displayed are supposed to be valid for "arbitrary distribution of linear dislocations". This includes wedge dislocations (also called disclinations in soft matter physics), which motivates the use of 3D geometry with curvature, but also screw dislocations which motivates the use of 3D geometry with torsion as well. Hence, in the strictest sense, it is the basics of Riemann-Cartan 3D gravity that should be recalled in section 2, with the spin connection and Cartan’s torsion tensor. For sake of clarity, the authors should clearly justify why only Riemann (curved) geometry is considered and why the tools of Riemann-Cartan geometry are not needed in the problem of they are investigating.

A second point that could improve the present paper is about the bibliography. At several steps, the authors emphasize that defects provide an interesting connection between relativistic gravity and elastic media. That is not the only interplay between these two fields and since the 1990’s several authors took that idea so seriously it has become an own research field: analogue gravity. The authors should mention recent review articles on defects in the context of analogue gravity in order to underline the importance of their work besides elasticity theory.  Moreover, mention should be made about the continuum theory of lattice defects pioneered by JD Eshelby (standard references in the mechanical engineering community) and how the present approach enriches it.

These points have to be adressed before considering the paper for publication in Universe.   

[1] R. Puntigam and H. H. Soleng, Class. Quant. Grav. 14, 1129 (1997).

Author Response

Sure, dislocations are related to torsion. Therefore we added the first paragraph in section 2:

"In the geometric theory of defects, dislocations and disclinations correspond to
nontrivial torsion and curvature, respectively. If disclinations are absent,
then curvature vanishes, and we are left with teleparallel space. It is well
known that teleparallel gravity is equivalent to general relativity (see e.g.
[45]. So, we shall use general relativity as more familiar
background. In this way, we obtain metric as the solution of Euclidean three
dimensional Einstein equations. Afterwards, the vielbein field and torsion can be
reconstructed but this is not needed in the case under consideration."

Moreover we added references [13,14] connected to this issue.

About analogue gravity. We added two references and the paragraph in the introduction:

"Last years great attention is paid to analogue gravity [39] which
investigates analogues of gravitational field phenomena in condensed matter
physics. In particular, behavior of fields in curved spacetime can be studied in
the laboratory (see [40] for review). In analogue gravity, nontrivial
metrics arise as the consequence of field equations for condensed matter which
differ from gravity equations. On the contrary, in the geometric theory of
defects, gravity equations themselves are used for description of defects
distribution in matter."

Coonection to traditional approach. We added the reference to Eshelby's paper and two paragraphs in the introduction:

"The traditional theory of single defects within the elasticity theory
[4] uses the displacement vector field as the main independent
variable with appropriate boundary conditions. In the geometric theory of
single dislocations, the only independent variable is the vielbein field satisfying
second order equilibrium equations. The displacement vector field is
reconstructed afterwards from the triad field in those domains where geometry is
Euclidean (curvature and torsion vanish). In the present paper, we follow
exactly this way of thought. First we solve three dimensional Einstein equations
and find the metric. Then we find the transformation of coordinates which brings
the obtained solution to the Euclidean metric outside the dislocation sources,
the displacement vector field being defined by the coordinate transformation.

It is understood that for description of continuous distribution of defects one
has to introduce new variable instead of the displacement vector field
[1-5] which is, in fact, the triad or
metric field. The geometric theory of defects proposes equations for this new
variable based on Euclidean three dimensional gravity models in
Riemann--Cartan spacetimes."

Also we added the last paragraph to section 4 expaining how to compute the displacement vecor field.

Reviewer 2 Report

Comments and Suggestions for Authors

The manuscript explores the geometric theory of defects in elastic media, treating it as a manifold with nontrivial Riemann–Cartan geometry. Specifically, it delves into the solution of three-dimensional Euclidean general relativity equations incorporating linear parallel sources, depicting elastic media with combined wedge and screw dislocations. The research is a valuable contribution to the understanding of the distribution of defects within elastic materials.

Strengths:

  1. The paper successfully formulates the equilibrium equations for vielbein in the context of three-dimensional general relativity with linear sources, providing a novel solution. This solution is analogous to the Lorentzian version and describes a diverse distribution of parallel combined wedge and screw dislocations in elastic media.

  2. The conclusion effectively summarizes the findings, emphasizing the significance of the obtained metric in describing the distribution of elastic stresses within media. The bridge to experiments through the use of an elastic gauge is a thoughtful suggestion for future studies.

Recommendations:

  1. Clarification of Applications: The authors assert the importance of understanding elastic media with dislocations for applications. To enhance the interest of a general audience, it would be beneficial if the authors could provide concrete examples of these applications and support their claims with references. This would make the paper more accessible to a wider audience and underscore the practical implications of the research.

  2. Clarity on Equation (46): In Equation (46), there is a mention of the constant θ parameter, but it also appears as dθ. This might cause confusion. A clarification is needed to ensure that the notation is consistent and clear. The question of whether it should be dθ or dφ needs to be addressed, as it affects the interpretation of the equation.

  3. Update of References: The references cited in the manuscript regarding the geometric theory of defects appear to be dated. It is recommended that the authors incorporate more recent references to highlight the current state of research in this field. This will not only demonstrate the relevance of the work but also provide readers with access to the latest developments in the geometric theory of defects.

Conclusion: The manuscript presents a valuable contribution to the geometric theory of defects in elastic media. Addressing the recommendations outlined above would significantly enhance the clarity, relevance, and overall appeal of the paper, making it a stronger candidate for acceptance in the Universe.

Author Response

1. We added the paragraph in the introduction:

"We are not aware of experimental confirmation of the geometric theory of defects, but there are some attempts to use these ideas in explaining known properties of real solids (see, for example, [41, 42, 43]).

2. The misprint is corrected.

3. References [13,14,16,34,41,42,43] are added.

Reviewer 3 Report

Comments and Suggestions for Authors

The authors obtain a solution, in the geometric theory of defects, for a configuration with an arbitrary number of linear parallel sources that describe combined wedge and screw dislocations. This is an interesting paper, with careful calculations and detailed explanations on how this solution is achieved. The only drawback is that, in order that the calculation be feasible, the authors must do it in a gauge for which the connection to a particular elastic medium configuration is elusive. As the authors mention, this is a technical problem that one should be able to solve in the course of time.

All things considered, my opinion is that this is a good paper, with results that are sound and worth being communicated. I therefore recommend its publication in this special issue of Universe.

Author Response

We agree that it is important to rewrite the obtained solution in the gauge related to particular elestic media. Unfortunately, this technical problem requires a lot of calculation. We hope to adress this issue in future publication.

Round 2

Reviewer 1 Report

Comments and Suggestions for Authors

All the problems have been addressed but one: the bibliography. For instance, [40] is indeed not a recent reference on analogue gravity (by the way, defects are hardly mentioned in it). More recent reviews are:

Jacquet, M. J., Weinfurtner, S., & König, F. (2020). The next generation of analogue gravity experiments. Philosophical Transactions of the Royal Society A378(2177), 20190239.

Fumeron, S., Berche, B., & Moraes, F. (2021). Geometric theory of topological defects: methodological developments and new trends. Liquid Crystals Reviews9(2), 85-110.

After fixing this last point, the paper will be suitable for publication in MDPI Universe.  

Author Response

References updated.